# Surrogate Regret Bounds for Polyhedral Losses

**Rafael Frongillo**
University of Colorado Boulder
raf@colorado.edu

**Bo Waggoner**
University of Colorado Boulder
bwag@colorado.edu

## Abstract

Surrogate risk minimization is an ubiquitous paradigm in supervised machine learning, wherein a target problem is solved by minimizing a surrogate loss on a dataset. Surrogate regret bounds, also called excess risk bounds, are a common tool to prove generalization rates for surrogate risk minimization. While surrogate regret bounds have been developed for certain classes of loss functions, such as proper losses, general results are relatively sparse. We provide two general results. The first gives a linear surrogate regret bound for any polyhedral (piecewise-linear and convex) surrogate, meaning that surrogate generalization rates translate directly to target rates. The second shows that for sufficiently non-polyhedral surrogates, the regret bound is a square root, meaning fast surrogate generalization rates translate to slow rates for the target. Together, these results suggest polyhedral surrogates are optimal in many cases.

## 1 Introduction

In supervised learning, our goal is to learn a hypothesis that solves some target problem given labeled data. The problem is often specified by a target loss $\ell$ that is discrete in the sense that the prediction lies in a finite set, as with classification, ranking, and structured prediction. As minimizing the target $\ell$ over a data set is typically NP-hard, we instead turn to the general paradigm of *surrogate risk minimization*. Here a surrogate loss function $L$ is chosen and minimized instead of the target $\ell$. A link function $\psi$ then maps the surrogate predictions to target predictions.

Naturally, a central question in surrogate risk minimization is how to choose the surrogate $L$. A minimal requirement is statistical *consistency*, which roughly says that approaching the best possible surrogate loss will eventually lead to the best possible target loss. A more precise goal is to understand the rate at which this target minimization occurs, called the *generalization rate*: how quickly the target loss approaches the best possible, in terms of the number of data points. In other words, if we define the *regret* of a hypothesis to be its performance relative to the Bayes optimal hypothesis, we wish to know how quickly the $\ell$-regret approaches zero.

Surrogate regret bounds are a common tool to prove generalization rates for surrogate risk minimization, by bounding the target regret as a function of the surrogate regret. This bound allows generalization rates for surrogate losses—of which there are many, often leveraging properties of $L$ like strong convexity and smoothness—to imply rates for the target problem. Roughly speaking, a surrogate regret bound takes the form

$$\forall \text{ surrogate hypotheses } h, \quad \operatorname{Regret}_\ell(\psi \circ h) \leq \zeta(\operatorname{Regret}_L(h)), \tag{1}$$

where the *regret transfer function* $\zeta : \mathbb{R}_+ \to \mathbb{R}_+$ controls how surrogate regret translates to target regret. For example, if we have a sequence of surrogate hypotheses $h_n$ achieving a fast rate $\operatorname{Regret}_L(h_n) = O(1/n)$, and we have a regret transfer of $\zeta(\epsilon) = \sqrt{\epsilon}$, then we can only guarantee a slower target rate of $\operatorname{Regret}_\ell(\psi \circ h_n) = O(1/\sqrt{n})$. The ideal regret transfer is *linear*, i.e. $\zeta(\epsilon) = C \cdot \epsilon$ for some $C > 0$, which would have directly translated the fast-rate guarantee to the target.

35th Conference on Neural Information Processing Systems (NeurIPS 2021).

Much is known about surrogate regret bounds for particular problems, especially for binary classification and related problems like bipartite ranking; see § 1.1 for a discussion of the literature. General results which span multiple target problems, however, are sparse. Furthermore, we still lack guidance as to which surrogate will lead to the best overall target generalization rate, as this rate depends on both the surrogate generalization rate and the regret transfer function. For example, consider the choice between a *polyhedral* (piecewise-linear and convex) surrogate, like hinge loss, or a smooth surrogate like exponential or squared loss. The surrogate rate for polyhedral losses will likely be a slower $O(1/\sqrt{n})$, whereas a faster $O(1/n)$ rate is typical for smooth surrogates—but what is the tradeoff when it comes to the regret transfer function $\zeta$, which ultimately decides the target generalization rate? To answer these types of questions, we need more general results.

Our first result is a general surrogate regret bound for polyhedral surrogates. Perhaps surprisingly, we show that *every* consistent surrogate in this class achieves a linear regret transfer, meaning their surrogate generalization rates translate directly to target rates, up to a constant factor.

**Theorem 1** (Linear bound for polyhedral). *Let $L$ be any polyhedral surrogate and $\psi$ any link function such that $(L, \psi)$ is consistent for the target loss $\ell$. Then $L$ satisfies Equation 1 with a linear $\zeta$.*

The recent embedding framework of Finocchiaro et al. [10] shows how to construct consistent polyhedral surrogates for any discrete target loss; Theorem 1 additionally endows these surrogates with linear regret bounds. The same framework shows that every polyhedral loss is a consistent surrogate for some discrete target. Theorem 1 therefore immediately applies to the Weston–Watkins surrogate [31], Lovász hinge [10, 33], and other polyhedral surrogates in the literature currently lacking regret bounds.[1]

We also give a partial converse: If the surrogate is sufficiently "non-polyhedral", i.e. smooth and locally strongly convex, then $\zeta$ cannot shrink faster than a square root. For these losses, therefore, even fast surrogate generalization rates would likely translate to slow rates for the target.

**Theorem 2** (Lower bound for "non-polyhedral"). *Let $L$ be a locally strongly convex surrogate with locally Lipschitz gradient. If $L$ satisfies Equation 1 for a target loss $\ell$, then there exists $c > 0$ such that, for all small enough $\epsilon \geq 0$, we have $\zeta(\epsilon) \geq c \cdot \sqrt{\epsilon}$.*

While it may seem intuitive that $\zeta = \sqrt{\cdot}$ for strongly convex losses, it is well-known that losses like exponential loss for binary classification also have $\zeta = \sqrt{\cdot}$, even though they are neither strongly convex nor strongly smooth [4]. Theorem 2 offers an explanation for this phenomenon, and together with Theorem 1 clarifies an intriguing trend in the literature: $\zeta$ is typically either linear or square-root.

Taken together, our results shed light on the dichotomy expressed above: slower polyhedral surrogate rates of $O(1/\sqrt{n})$ will translate to the target unchanged, while most smooth surrogates, even if achieving a fast surrogate generalization rate of $O(1/n)$, will likely yield the same target rate of $O(1/\sqrt{n})$. Our results therefore suggest an inherent tradeoff fast surrogate rates and optimal regret transfer, a phenomenon also observed by Mahdavi et al. [17] for binary classification. In particular, our results suggest that polyhedral losses may achieve an overall generalization rate to the target problem with the optimal dependence on the number of data points.

The proof of our upper bound, Theorem 1, is nonconstructive. After proving the main results (Sections 3 and 4), we make it constructive in Section 5. Specifically, the regret transfer function for a polyhedral loss can be bounded by $\zeta(t) \leq \alpha \cdot t$ where the constant is of the form $\alpha = \alpha_\ell \cdot \alpha_L \cdot \alpha_\psi$. Each component of $\alpha$ depends only on its respective object $\ell$, $L$, and $\psi$. A corollary of the derivation is that a polyhedral loss $L$ and link $\psi$ are consistent for $\ell$ if and only if $\psi$ is $\epsilon$-*separated*, a condition introduced in Finocchiaro et al. [10].

## 1.1 Literature on surrogate regret bounds

Surrogate regret bounds were perhaps developed first by Zhang [37], as a method to prove statistical consistency for various surrogate losses for binary classification. These results were then generalized by Bartlett et al. [4] for margin losses, and Reid and Williamson [26] for composite proper losses.

---

[1]Many of these polyhedral surrogates mentioned are actually known to be *inconsistent* for their intended target problems. Nonetheless, the embedding framework of Finocchiaro et al. [10] shows that any polyhedral surrogate is consistent for *some* target problem, namely the one that it embeds. It is to this embedded target loss function, rather than the intended target, that our linear regret bounds apply.

These latter works and others give a fairly complete picture for binary classification: a characterization of calibrated surrogates, and precise closed forms for the optimal $\zeta$ for each. By extension, close variations on binary classification are also well-understood, such as cost-sensitive binary classification [28], binary classification with an abstain/reject option [3, 8, 34], bipartite ranking [1, 13, 18], and ordinal regression via reduction to binary classification [22]. For multiclass classification, we have nice consistency characterizations with respect to 0-1 loss [30], with some surrogate regret bounds [9, 23].

Beyond binary classification and 0-1 loss, our knowledge drops off precipitously, with few general results. There are some works which provide partial understand for specific target problems. For example, in the problem of multiclass classification with an abstain option, Ramaswamy et al. [25] give three consistent polyhedral surrogates (two of which are commonly used as inconsistent surrogates for 0-1 loss), along with linear surrogate regret bounds. (See also Ni et al. [19, Theorems 7 & 8] and Charoenphakdee et al. [6, Theorem 8] for other surrogate regret bounds for the same problem.) Ramaswamy et al. [24] show how to extend these surrogates to nested, hierarchical versions, again providing linear regret bounds. Motivated by these polyhedral surrogates, together with examples arising in top-$k$ prediction [16, 32] and other structured prediction problems [33], Finocchiaro et al. [10] introduce an embedding framework for polyhedral surrogates. This framework shows that all polyhedral surrogates are consistent for some discrete target problem, and vice versa, but do not provide surrogate regret bounds. Our results fill this gap, giving linear regret bounds for all polyhedral losses, and thus all discrete target problems.

Perhaps closest in spirit to our work are several recent works on general structured prediction problems. Ciliberto et al. [7, Theorem 2] and Osokin et al. [21, Theorem 7] give surrogate regret bounds for a quadratic surrogate, for a general class of structed prediction problems. Blondel [5, Proposition 5] gives surrogate regret bounds for a tighter class of quadratic surrogates based on projection oracles. Nowak-Vila et al. [20, Theorem 4.4] give even more general bounds, using a broader class of surrogates based on estimating a vector-valued expectation. All of these works apply to a broad range of discrete target problems. The main difference to our work is that their upper bounds are all for strongly convex surrogates,[2] whereas ours are for polyhedral surrogates. Osokin et al. [21, Theorem 8] and Nowak-Vila et al. [20, Theorem 4.5] also provide lower bounds, in the same spirit as our Theorem 2 but only for the classes of surrogates they study. In particular, our results recover their bounds up to a constant factor.

Finally, in the binary classification setting, some works have used surrogate regret bounds to study the relationship between generalization *rates* for the surrogate and target, as discussed above. Zhang et al. [36] give detailed results about how surrogate regret bounds actually convert the surrogate rate to the target rate, by carefully analyzing the behavior of the regret tranfer function near 0. They argue that hinge loss achieves strictly better rates than smooth surrogates like logistic and exponential loss; notably, though, this claim implicitly assumes that generalization rate is the same for all surrogates under consideration. By contrast, Mahdavi et al. [17], as we do, consider that the surrogate generalization rate may *depend* on the choice of surrogate. They study a particular family of smoothed hinge losses, and show a direct tradeoff: smoother losses yield faster surrogate generalization rates, but "slower" regret transfers. Without placing further assumptions on the data, therefore, Mahdavi et al. [17] observe the same general phenomenon as we do: while smoothness may help surrogate generalization rates, it can hurt the regret transfer just as much. In particular, without further assumptions, polyhedral surrogates appear to achieve optimal overall generalization rates.

## 2 Preliminaries

**Target problems.** We consider supervised machine learning problems where data points have features in $\mathcal{X}$ and labels in $\mathcal{Y}$, a finite set. The *target problem* is to learn a hypothesis mapping $\mathcal{X}$ to a *report* (or *prediction*) *space* $\mathcal{R}$, possibly different from $\mathcal{Y}$. For example, binary classification has $\mathcal{Y} = \mathcal{R} = \{-1, +1\}$, whereas for ranking problems $\mathcal{R}$ may be the set of permutations of $\mathcal{Y}$. In this paper, we assume $\mathcal{R}$ is a finite set. The target problem is specified by the *target loss* $\ell : \mathcal{R} \to \mathbb{R}_+^{\mathcal{Y}}$, which maps a report (prediction) $r \in \mathcal{R}$ to the vector $\ell(r)$ of nonnegative loss values for each label,

---

[2]These works express surrogate regret bounds with $\zeta^{-1}$ appearing on the left-hand side of eq. (1), and thus the terms "upper bound" and "lower bound" are reversed from their terminology.

i.e. $\ell(r) = (\ell(r)_y)_{y \in \mathcal{Y}}$. Given a hypothesis $g : \mathcal{X} \to \mathcal{R}$ and a data point $(x, y) \in \mathcal{X} \times \mathcal{Y}$, the loss is $\ell(g(x))_y$. We make a minimal non-redundancy assumption on $\ell$, formalized in Section 2.1. We may refer to $\ell$ as a *discrete* target loss to emphasize our assumption that $\mathcal{R}$ and $\mathcal{Y}$ are finite.

**Surrogate problems.** We recall the surrogate risk minimization framework. First, one constructs a *surrogate loss function* $L : \mathbb{R}^d \to \mathbb{R}_+^{\mathcal{Y}}$. Then, one minimizes it to learn a hypothesis of the form $h : \mathcal{X} \to \mathbb{R}^d$. Finally, one applies a *link function* $\psi : \mathbb{R}^d \to \mathcal{R}$, which maps $h$ to the implied target hypothesis $\psi \circ h$.[3] In other words, if a surrogate prediction $h(x) \in \mathbb{R}^d$ implies a prediction $\psi(h(x)) \in \mathcal{R}$ for the target problem. For example, in binary classification, the target problem is given by 0-1 loss, $\ell_{\mathrm{zo}}(r)_y = \mathbb{1}\{r \neq y\}$, with surrogate losses typically defined over $\mathbb{R}$. Common surrogate losses for 0-1 loss include hinge loss $L_{\mathrm{hinge}}(u)_y = \max(1 - uy, 0)$, logistic loss $L_{\mathrm{log}}(u)_y = \ln(1 + \exp(-yu))$, and exponential loss $L_{\mathrm{exp}}(u)_y = \exp(-yu)$. These surrogates are typically associated with the link $\psi : u \mapsto \mathrm{sign}(u)$, which maps negative predictions to $-1$ and nonnegative predictions to $+1$, breaking ties arbitrarily at 0.

**Surrogate regret bounds.** The suitability and efficiency of a chosen surrogate is often quantified with a *surrogate regret bound*. Formally, the *surrogate regret* of $h$ with respect to data distribution $\mathcal{D}$ is given by $R_L(h; \mathcal{D}) = \mathbb{E}_{(X,Y) \sim \mathcal{D}} L(h(X))_Y - \inf_{h' : \mathcal{X} \to \mathbb{R}^d} \mathbb{E}_{(X,Y) \sim \mathcal{D}} L(h'(X))_Y$, where the minimum is taken over all measurable functions. (One may equivalently assume the Bayes optimal hypothesis is in the function class.) Given $h$, the *target regret* of the implied hypothesis $\psi \circ h$ is $R_\ell(\psi \circ h; \mathcal{D}) = \mathbb{E}_{(X,Y) \sim \mathcal{D}} \ell(\psi(h(X)))_Y - \inf_{h' : \mathcal{X} \to \mathcal{R}} \mathbb{E}_{(X,Y) \sim \mathcal{D}} \ell(h'(X))_Y$, where we recall that $\psi$ is the link function.

A surrogate regret bound answers the following question: when we minimize the surrogate regret at a rate $f(n)$, given $n$ data points, how fast does the target regret diminish? Formally, we say that $\zeta : \mathbb{R}_+ \to \mathbb{R}_+$ is a *regret transfer function* if $\zeta$ is continuous at 0 and satisfies $\zeta(0) = 0$. Given a rerget transfer function $\zeta$, we say that $(L, \psi)$ guarantees a *regret transfer of $\zeta$* for $\ell$ if

$$\forall \mathcal{D}, \forall h : \mathcal{X} \to \mathbb{R}^d, \quad R_\ell(\psi \circ h; \mathcal{D}) \leq \zeta(R_L(h; \mathcal{D})). \tag{2}$$

We note that a classic minimal requirement for $(L, \psi)$ is that they be *consistent* for $\ell$, which is the case if there exists any regret transfer $\zeta$ guaranteed by $(L, \psi)$ for $\ell$ [29]. In other words, consistency implies that $R_L(h; \mathcal{D}) \to 0 \implies R_\ell(\psi \circ h; \mathcal{D}) \to 0$, but says nothing about the relative rates.

**Polyhedral surrogates.** A function $f : \mathbb{R}^d \to \mathbb{R}$ is called *polyhedral* if it can be written as a pointwise maximum of a finite set of affine functions [27, § 19]. The epigraph of a polyhedral function is a polyhedral set, i.e. the intersection of a finite set of closed halfspaces; thus polyhedral functions are always convex. We say a surrogate loss $L : \mathbb{R}^d \to \mathbb{R}_+^{\mathcal{Y}}$ is *polyhedral* if, for each fixed $y \in \mathcal{Y}$, the loss as a function of prediction, i.e. $r \mapsto L(r)_y$, is polyhedral.

## 2.1 Property elicitation and the conditional approach

Our technical approach is based on property elicitation, which studies the structure of $\ell$ and $L$ as they relate to $\Delta_{\mathcal{Y}}$. This involves a *conditional* perspective. Given a data distribution $\mathcal{D}$ on $\mathcal{X} \times \mathcal{Y}$, for $(X, Y) \sim \mathcal{D}$, we consider the format of possible target predictions $g(x) = r \in \mathcal{R}$; surrogate predictions $h(x) = u \in \mathbb{R}^d$; and conditional distributions $\Pr[\cdot \mid X] = p \in \Delta_{\mathcal{Y}}$. By dropping the role of the hypotheses and the feature space $\mathcal{X}$, we can focus on the relationships of the functions $\ell(r)_y$ and $L(u)_y$ and the conditional distribution $p$.

In particular, in our notation the expected $L$-loss of prediction $u$ on distribution $p$ is simply the inner product $\langle p, L(u) \rangle$. The *Bayes risk* of $L$ is $\underline{L}(p) = \inf_{u \in \mathbb{R}^d} \langle p, L(u) \rangle$, and similarly the Bayes risk of $\ell$ is $\underline{\ell}(p) = \inf_{r \in \mathcal{R}} \langle p, \ell(r) \rangle$. We abuse notation to define the *conditional surrogate regret* $R_L(u, p) = \langle p, L(u) \rangle - \underline{L}(p)$, and similarly the *conditional target regret* $R_\ell(r, p) = \langle p, \ell(r) \rangle - \underline{\ell}(p)$. Given a regret transfer function $\zeta$, we say $(L, \psi)$ guarantees a *conditional regret transfer* of $\zeta$ for $\ell$ if, for all $p \in \Delta_{\mathcal{Y}}$ and all $u \in \mathbb{R}^d$, $R_\ell(\psi(u), p) \leq \zeta(R_L(u, p))$.

The following is a direct result of Jensen's inequality, using that $R_L(h; \mathcal{D}) = \mathbb{E}_X R_L(h(X), p_X)$ where $p_X$ is the conditional distribution on $Y$ given $X$.

---

[3]The term "link function" can carry a more specific meaning in the statistics literature, for example when discussing generalized linear models. Our usage here is more general: any map from surrogate reports to target reports is a valid link function, though perhaps not a calibrated one.

**Observation 1.** *If $(L, \psi)$ guarantee a conditional regret transfer of $\zeta$ for $\ell$, and $\zeta$ is concave, then $(L, \psi)$ guarantee a regret transfer of $\zeta$ for $\ell$.*

**Property elicitation.** To begin, we define a *property*, which is essentially a statistic or summary of a distribution, such as the mode. We then define what it means for a loss function to *elicit* a property. We write $\Gamma : \Delta_{\mathcal{Y}} \rightrightarrows R$ as shorthand for $\Gamma : \Delta_{\mathcal{Y}} \to 2^R \setminus \{\emptyset\}$.

**Definition 1** (Property, level set). *A property is a function $\Gamma : \Delta_{\mathcal{Y}} \rightrightarrows R$, for some set $R$. The level set of $\Gamma$ for report $r$ is the set $\Gamma_r = \{p \in \Delta_{\mathcal{Y}} : r \in \Gamma(p)\}$.*

The following definition applies both to target losses and surrogate losses.

**Definition 2** (Elicits, report space). *A loss function $f : R \to \mathbb{R}_+^{\mathcal{Y}}$, where the domain $R$ is called the report space, elicits a property $\Gamma : \Delta_{\mathcal{Y}} \rightrightarrows R$ if*

$$\forall p \in \Delta_{\mathcal{Y}}, \quad \Gamma(p) = \underset{r \in R}{\arg\min} \langle p, f(r) \rangle =: \mathrm{prop}[f](p) . \tag{3}$$

As each loss $f$ elicits a unique property, we refer to it as $\mathrm{prop}[f]$. In particular, we will consistently use the notation $\Gamma = \mathrm{prop}[L]$ where $L$ is a surrogate loss with report space $\mathbb{R}^d$, and we will use $\gamma = \mathrm{prop}[\ell]$ where $\ell$ is a target loss with finite report space $\mathcal{R}$. For example, $\mathrm{prop}[\ell_{\mathrm{zo}}]$ is simply the mode of $p$, given by $\mathrm{mode}(p) = \{-1, 1\}$ if $p = 1/2$, and $\mathrm{mode}(p) = \{\mathrm{sign}(2p - 1)\}$ otherwise.

We will assume throughout that the target loss $\ell : \mathcal{R} \to \mathbb{R}_+^{\mathcal{Y}}$ is *non-redundant*, meaning that all target reports $r \in \mathcal{R}$ are possibly uniquely optimal. Formally, letting $\gamma = \mathrm{prop}[\ell]$, we suppose that for all $r \in \mathcal{R}$, there exists $p \in \Delta_{\mathcal{Y}}$ with $\gamma(p) = \{r\}$.[4]

**Calibration and consistency.** It is known that consistency implies the weaker condition of *calibration*, which implies the still-weaker, but very useful, condition of *indirect elicitation*. To define these conditions, consider $L, \psi, \ell$ and $\gamma = \mathrm{prop}[\ell]$. Recall that $\gamma(p)$ is the set of target reports that are optimal for $p$. The typical definition of indirect elicitation, which we reformulate below, is as follows: a surrogate-link pair $(L, \psi)$ *indirectly elicits* $\gamma$ if $u \in \mathrm{prop}[L](p) \implies \psi(u) \in \gamma(p)$ for all $p \in \Delta_{\mathcal{Y}}$. Note that if $\psi(u) \notin \gamma(p)$, then $u$ is a "bad" (suboptimal) surrogate prediction for $p$. Let $B_{L,\psi,\ell}(p) = \{R_L(u, p) : \psi(u) \notin \gamma(p)\}$, the set of expected surrogate losses resulting from bad predictions. Thus, we may equivalently state indirect elicitation as the requirement that suboptimal surrogate reports be bad, which clarifies the relationship to calibration: calibration further requires a nonzero gap between the expected loss of a "good" report and that of any bad one.

**Definition 3** (Indirectly Elicits, Calibrated). *Let $\ell$ be a target loss and $\gamma = \mathrm{prop}[\ell]$. A surrogate-link pair $(L, \psi)$ indirectly elicits $\gamma$ if $0 \notin B_{L,\psi,\ell}(p)$ for all $p \in \Delta_{\mathcal{Y}}$. The pair $(L, \psi)$ is calibrated for $\ell$ if $0 < \inf B_{L,\psi,\ell}(p)$ for all $p \in \Delta_{\mathcal{Y}}$.*

**Fact 1** ([4, 29]). *Let $\ell : \mathcal{R} \to \mathbb{R}_+^{\mathcal{Y}}$ be a target loss with finite $\mathcal{R}$ and $\mathcal{Y}$. If a surrogate and link $(L, \psi)$ are consistent for $\ell$, then $(L, \psi)$ are calibrated for $L$. If $(L, \psi)$ are calibrated for $\ell$, then they indirectly elicit $\ell$.*

For example, it is well-known that hinge loss $L_{\mathrm{hinge}}$ is calibrated and consistent for 0-1 loss $\ell_{\mathrm{zo}}$ using the link $\psi : u \mapsto \mathrm{sign}(u)$. Recall that $\ell_{\mathrm{zo}}$ elicits the mode. One can verify that indeed $(L_{\mathrm{hinge}}, \psi)$ indirectly elicit the mode; for example, $B_{L_{\mathrm{hinge}}, \psi, \ell_{\mathrm{zo}}}(0) = B_{L_{\mathrm{hinge}}, \psi, \ell_{\mathrm{zo}}}(1) = [1, \infty)$ and $B_{L_{\mathrm{hinge}}, \psi, \ell_{\mathrm{zo}}}(1/2) = \emptyset$, neither of which contain 0.

## 3 Upper Bound: Polyhedral Surrogates

In this section, we describe the proof of Theorem 1, that polyhedral surrogates guarantee linear regret transfers. At a high level, the proof (whose details appear in Appendix A) has two parts:

1. Fix $p \in \Delta_{\mathcal{Y}}$. Prove a regret bound for this fixed $p$, namely a constant $\alpha_p$ such that $R_\ell(\psi(u), p) \leq \alpha_p R_L(u, p)$ for all $u \in \mathbb{R}^d$.

2. Apply this regret bound to each $p$ in a carefully chosen finite subset of $\Delta_{\mathcal{Y}}$. Argue that the maximum $\alpha_p$ from this finite set suffices for the overal regret bound.

---

[4]Technically, this definition differs from the one given in literature [10, 11], but the two are equivalent for finite elicitable properties.

The first part will actually hold for any calibrated surrogate. The second part relies on a crucial fact about polyhedral losses $L$: their properties $\Gamma = \mathrm{prop}[L]$ cover the whole simplex with only a finite number of polyhedral level sets [10]. Thus, although the prediction space $\mathbb{R}^d$ is infinite, one can restrict to just a finite set of prediction points and always have a global minimizer of expected surrogate loss.

## 3.1 Linear transfer for fixed $p$

First, we establish a linear transfer function for any fixed conditional distribution $p$. This statement holds for any calibrated surrogate, not necessarily polyhedral.

**Lemma 1.** *Let $\ell : \mathcal{R} \to \mathbb{R}_+^{\mathcal{Y}}$ be a discrete target loss and suppose the surrogate $L : \mathbb{R}^d \to \mathbb{R}_+^{\mathcal{Y}}$ and link $\psi : \mathbb{R}^d \to \mathcal{R}$ are calibrated for $\ell$. Then for any $p \in \Delta_{\mathcal{Y}}$, there exists $\alpha_p \geq 0$ such that, for all $u \in \mathbb{R}^d$,*

$$R_\ell(\psi(u), p) \leq \alpha_p R_L(u, p).$$

While useful, this result is somewhat nonconstructive and potentially loose. In Section 5, we unpack the constant $\alpha_p$ for polyhedral losses in particular.

## 3.2 Linear overall transfer function

Given Lemma 1, we might hope to obtain that, for any calibrated surrogate, there exists $\alpha = \sup_p \alpha_p$ such that $R_\ell(\psi(u), p') \leq \alpha R_L(u, p')$ for all $p'$. However, we know this is false in general: not all surrogates yield linear regret transfers! Indeed, for many surrogates, the supremum will be $+\infty$.

However, polyhedral surrogates have a special structure that allows us to show $\alpha$ is finite. We first present some tools that hold for general surrogates, then the key implication from $L$ being polyhedral.

**Lemma 2** ([12]). *If $(L, \psi)$ indirectly elicits $\gamma$, then $\Gamma = \mathrm{prop}[L]$ refines $\gamma$ in the sense that, for all $u \in \mathbb{R}^d$, there exists $r \in \mathcal{R}$ such that $\Gamma_u \subseteq \gamma_r$.*

The next lemma states that surrogate regret is linear in $p$ on any fixed level set $\Gamma_{u^*}$ of $\Gamma = \mathrm{prop}[L]$. By combining this fact with Lemma 2, we obtain that target regret is linear on these level sets as well.

**Lemma 3.** *Suppose $(L, \psi)$ indirectly elicits $\gamma = \mathrm{prop}[\ell]$ and let $\Gamma = \mathrm{prop}[L]$. Then for any fixed $u, u^* \in \mathbb{R}^d$ and $r \in \mathcal{R}$, the functions $R_L(u, \cdot)$ and $R_\ell(r, \cdot)$ are linear on $\Gamma_{u^*}$ in their second arguments.*

A key fact we use about polyhedral losses is that they have a finite set of optimal sets.

**Lemma 4.** *If $L : \mathbb{R}^d \to \mathbb{R}_+^{\mathcal{Y}}$ is polyhedral, then $\Gamma = \mathrm{prop}[L]$ has a finite set of level sets that union to $\Delta_{\mathcal{Y}}$. Moreover, these level sets are polytopes.*

We are now ready to restate the main upper bound and sketch a proof.

**Theorem 3** (Theorem 1 restated). *Suppose the surrogate loss $L : \mathbb{R}^d \to \mathbb{R}_+^{\mathcal{Y}}$ and link $\psi : \mathbb{R}^d \to \mathcal{R}$ are consistent for the target loss $\ell : \mathcal{R} \to \mathbb{R}_+^{\mathcal{Y}}$. If $L$ is polyhedral, then $(L, \psi)$ guarantee a linear regret transfer for $\ell$, i.e. there exists $\alpha \geq 0$ such that, for all $\mathcal{D}$ and all measurable $h : \mathcal{X} \to \mathbb{R}^d$,*

$$R_\ell(\psi \circ h; \mathcal{D}) \leq \alpha R_L(h; \mathcal{D}).$$

The key point of the proof is that, by Lemma 4, the polyhedral loss $L$ has a finite set $U \subset \mathbb{R}^d$ of predictions such that (a) for each $u \in U$, the level set $\Gamma_u$ is a polytope, and (b) $\cup_{u \in U} \Gamma_u = \Delta_{\mathcal{Y}}$. Let $\mathcal{Q}$ be the finite set of all vertices of all these level sets. For any vertex $q \in \mathcal{Q}$, we have a linear regret transfer for this fixed $q$ with constant $\alpha_q$ from Lemma 1. Lemma 3 allows us to write the regret of a general $p$ as a convex combination of the regrets of its containing level set's vertices. So we obtain a bound of $\alpha = \max_{q \in \mathcal{Q}} \alpha_q$.

# 4 Lower bound

In this section, we show that if a surrogate loss is sufficiently "non-polyhedral", then the best regret transfer it can achieve is $\zeta(\epsilon)$ on the order of $\sqrt{\epsilon}$. (Proofs are deferred to Appendix B.) Specifically, we consider a relaxation of $L$ being both strongly smooth and strongly convex, as we now formalize.

Let $\Gamma = \mathrm{prop}[L]$ and $\gamma = \mathrm{prop}[\ell]$. We will assume $L$ is strongly smooth everywhere and strongly convex around some *boundary report for $L, \ell$*, which we define to be a $u_0 \in \mathbb{R}^d$ such that, for some $r, r' \in \mathcal{R}$, there exists a distribution $p_0$ such that $p_0 \in \Gamma_{u_0} \cap \gamma_r \cap \gamma_{r'}$. For the conditional distribution $p_0$, the surrogate report $u_0$ minimizes expected $L$-loss, and it may link to either $r$ or $r'$, as either of these minimize expected $\ell$-loss. Recall that an *open neighborhood* of $u_0$ is an open set in $\mathbb{R}^d$ containing $u_0$.

**Assumption 1.** $L : \mathbb{R}^d \to \mathbb{R}_+^{\mathcal{Y}}$ is a surrogate and $\psi : \mathbb{R}^d \to \mathcal{R}$ a link, consistent with the discrete target $\ell : \mathcal{R} \to \mathbb{R}_+^{\mathcal{Y}}$, satisfying the following: (i) For all $y \in \mathcal{Y}$, the function $L(\cdot)_y$ is differentiable with a locally Lipschitz gradient.[5] (ii) For some $\alpha > 0$ and some boundary report $u_0 \in \mathbb{R}^d$ for $L, \ell$, the expected loss function $u \mapsto \langle p, L(u) \rangle$ is $\alpha$-strongly convex on an open neighborhood of $u_0$.

This assumption is quite weak. In particular, because $\ell$ has a finite report set $\mathcal{R}$, boundary reports essentially always exist whenever $(L, \psi)$ indirectly elicits $\ell$. In particular, as the level sets of $\gamma$ are closed, convex, and union to the simplex [10, 15], there must be some pair of reports $r, r' \in \mathcal{R}$ with $\gamma_r \cap \gamma_{r'} \neq \emptyset$. We can then take $p_0 \in \gamma_r \cap \gamma_{r'}$, and any $u_0 \in \Gamma(p_0)$. One caveat is that we may have $\Gamma(p_0) = \emptyset$. While some surrogates may fail to have a minimizer for some distributions $p$, this typically happens when $p$ is on the boundary of the simplex, and not generally on the boundary between level sets. For example, consider exponential loss $L_{\exp}(u)_y = \exp(-uy)$, where $\mathrm{prop}[L_{\exp}](0) = \mathrm{prop}[L_{\exp}](1) = \emptyset$, but we still have $\mathrm{prop}[L_{\exp}](1/2) = \{0\}$.

The following result is our main lower bound.

**Theorem 4** (Stronger version of Theorem 2.). *Suppose the surrogate loss $L$ and link $\psi$ satisfy a regret transfer of $\zeta$ for a target loss $\ell$. If $L$, $\psi$, and $\ell$ satisfy Assumption 1, then there exists $c > 0$ such that, for some $\epsilon^* > 0$, for all $0 \leq \epsilon < \epsilon^*$, $\zeta(\epsilon) \geq c\sqrt{\epsilon}$.*

The key idea of the proof is to fix a boundary report $u_0$ and consider a sequence of distributions $p_\lambda \to p_0$ as $\lambda \to 0$. Target regret $R_\ell(\psi(u_0), p_\lambda)$ shrinks linearly in $\lambda$, but surrogate regret $R_L(u_0, p_\lambda)$ shrinks quadratically. The latter holds roughly because expected surrogate loss is strongly convex in some neighborhood of $u_0$ and is strongly smooth elsewhere.[6]

Theorem 4 captures a wide range of surrogate losses beyond those that are strongly convex and strongly smooth. For example, exponential loss $L_{\exp}(u)_y = \exp(-uy)$ is not strongly convex, but does satisfy Assumption 1; for part (ii), the boundary report is $u_0 = 0$ as we saw above, and $L_{\exp}$ is $1/e$-strongly convex on $(-1, 1) \ni u_0$.

As another example, consider Huber loss for binary classification. Here $\mathcal{Y} = \{\pm 1\}$ and the prediction space is $\mathbb{R}$, and Huber loss can be defined by letting the hinge loss be $t = \max\{0, 1 - ry\}$ and setting $L(r)_y = t^2$ if $t \leq 2$, else $4(t - 1)$. Theorem 4 does capture Huber, but needs the strength of Assumption 1(ii) as Huber is strongly convex on $[-1, 1]$ but linear outside that interval.

## 5 Unpacking the constant for polyhedral losses

Our main upper bound for polyhedral losses, Theorem 1, is mostly nonconstructive: it says polyhedral surrogates have some $\alpha$ such that $R_\ell(\psi \circ h; \mathcal{D}) \leq \alpha \cdot R_L(h; \mathcal{D})$. In this section, we make the result constructive by unpacking the implications of consistency for polyhedral surrogates. Proofs appear in Appendix C.

### 5.1 Contribution from the target, surrogate, and link

We will give a bound on the constant of the form $\alpha \leq \alpha_\ell \cdot \alpha_L \cdot \alpha_\psi$, i.e. one component each relating to the structure of the target loss, surrogate loss, and link. First, we will derive $\alpha_L$ using the theory of *Hoffman constants* from linear optimization [14]. Then, we will show that *any* link for a consistent polyhedral loss must satisfy a condition called $\epsilon$-separation, introduced in [10]. We will set $\alpha_\psi = \frac{1}{\epsilon}$. With $\alpha_\ell$ simply an upper bound on the target loss, we will finally prove $\alpha \leq \alpha_\ell \cdot \alpha_L \cdot \alpha_\psi$.

---

[5] A map $g : A \to B$ is locally Lipschitz if for every $a \in A$ there is an open neighborhood $U$ of $a$ such that $g|_A$ is Lipschitz continuous.

[6] A tempting alternative is to fix a distribution $p_0$ and consider a sequence of predictions. We know from Lemma 1 that this approach will fail, however: for any consistent surrogate and any fixed $p_0$, there is a linear regret transfer if we restrict to $p_0$.

### 5.1.1 Hoffman constants

The intuition we explore here is the linear growth rate of a given polyhedral function as we move away from its optimal set. In particular, consider the expected loss $u \mapsto \langle p, L(u) \rangle$. Expected loss should grow linearly with distance from the optimal set, which happens to be $\Gamma(p)$. In particular, the worst growth rate is governed roughly by the smallest slope of the function in any direction. This is captured by the *Hoffman constant* of an associated system of linear inequalities. In Appendix C, we use Hoffman constants to obtain the following result, where we define $H_{L,p}$.

**Lemma 5** ([14]). *Let $L : \mathbb{R}^d \to \mathbb{R}_+^{\mathcal{Y}}$ be a polyhedral loss with $\Gamma = \mathrm{prop}[L]$. Then for any fixed $p$, there exists some smallest constant $H_{L,p} \geq 0$ such that $d_\infty(u, \Gamma(p)) \leq H_{L,p} R_L(u, p)$ for all $u \in \mathbb{R}^d$.*

To foreshadow the usefulness of $H_{L,p}$, recall the proof strategy of the upper bound: use Lemma 1 to obtain a constant $\alpha_p$ for each $p$; then carefully choose a finite set of $p$'s and take the maximum constant. In the same way, we will be able to set the "loss" constant $\alpha_L = \max_p H_{L,p}$. First, however, we will need to investigate separated links.

### 5.1.2 Separated link functions

A link is $\epsilon$-separated if, for all pairs of surrogate reports $u, u^*$ such that $u^*$ is optimal for the surrogate and $u$ links to a suboptimal target report, we have $\|u^* - u\|_\infty > \epsilon$. The definition was introduced by Finocchiaro et al. [10].

**Definition 4** (Separated Link). *Let properties $\Gamma : \Delta_{\mathcal{Y}} \rightrightarrows \mathbb{R}^d$ and $\gamma : \Delta_{\mathcal{Y}} \rightrightarrows \mathcal{R}$ be given. We say a link $\psi : \mathbb{R}^d \to \mathcal{R}$ is $\epsilon$-separated with respect to $\Gamma$ and $\gamma$ if for all $u \in \mathbb{R}^d$ with $\psi(u) \notin \gamma(p)$, we have $d_\infty(u, \Gamma(p)) > \epsilon$, where $d_\infty(u, A) \doteq \inf_{a \in A} \|u - a\|_\infty$. Similarly, we say $\psi$ is $\epsilon$-separated with respect to $L$ and $\ell$ if it is $\epsilon$-separated with respect to $\mathrm{prop}[L]$ and $\mathrm{prop}[\ell]$.*

Recall from the previous subsection that Hoffman constants allowed us to show that polyhedral expected losses grow linearly as we move away from the optimal set. Now, $\epsilon$-separated link functions allow us to guarantee that we move at least a constant distance from the optimal set before we start linking to an "incorrect" report $r' \in \mathcal{R}$. But when does a polyhedral loss have an associated $\epsilon$-separated link?

Finocchiaro et al. [10] showed that $\epsilon$-separated links for polyhedral surrogates are calibrated. We show that the converse is in fact true: every calibrated link is $\epsilon$-separated for some $\epsilon > 0$. The proof follows a similar argument to that of Tewari and Bartlett [30, Lemma 6].

**Lemma 6.** *Let polyhedral surrogate $L : \mathbb{R}^d \to \mathbb{R}_+^{\mathcal{Y}}$, discrete loss $\ell : \mathcal{R} \to \mathbb{R}_+^{\mathcal{Y}}$, and link $\psi : \mathbb{R}^d \to \mathcal{R}$ be given such that $(L, \psi)$ is calibrated with respect to $\ell$. Then there exists $\epsilon > 0$ such that $\psi$ is $\epsilon$-separated with respect to $\Gamma \doteq \mathrm{prop}[L]$ and $\gamma \doteq \mathrm{prop}[\ell]$.*

### 5.1.3 Combining the loss and link

We can now follow the general proof strategy of the main upper bound, but constructively. Given Lemma 6, we can make the following definition.

**Definition 5** ($\epsilon_\psi$). *If $(L, \psi)$ are calibrated for $\ell$, then let $\epsilon_\psi \doteq \sup\{\epsilon \ : \ \psi \text{ is } \epsilon\text{-separated}\}$.*

We will also need an upper bound $C_\ell = \max_{r,p} R_\ell(r, p)$ on the regret of $\ell$. Since $R_\ell$ is convex in $p$, in particular this maximum is achieved at a vertex of the probability simplex.

**Definition 6** ($C_\ell$). *Given discrete loss $\ell : \mathcal{R} \to \mathbb{R}_+^{\mathcal{Y}}$, define $C_\ell = \max_{r,r' \in \mathcal{R}, y \in \mathcal{Y}} \ell(r)_y - \ell(r')_y$.*

Recall that Lemma 4 gave that, if $L$ is polyhedral, then $\Gamma = \mathrm{prop}[L]$ has a finite set of full-dimensional level sets, each a polytope, that union to the simplex.

**Definition 7** ($H_L$). *Given a polyhedral surrogate loss $L : \mathbb{R}^d \to \mathbb{R}_+^{\mathcal{Y}}$, let $\mathcal{Q}$ be the set of all vertices of the full-dimensional level sets of $\Gamma = \mathrm{prop}[L]$, and define $H_L \doteq \max_{p \in \mathcal{Q}} H_{L,p}$.*

**Theorem 5** (Constructive linear transfer). *Let $\ell : \mathcal{R} \to \mathbb{R}_+^{\mathcal{Y}}$ be a discrete target loss, $L : \mathbb{R}^d \to \mathbb{R}_+^{\mathcal{Y}}$ be a polyhedral surrogate loss, and $\psi : \mathbb{R}^d \to \mathcal{R}$ a link function. If $(L, \psi)$ are consistent for $\ell$, then*

$$(\forall h, \mathcal{D}) \quad R_\ell(\psi \circ h; \mathcal{D}) \leq \frac{C_\ell H_L}{\epsilon_\psi} R_L(h; \mathcal{D}) .$$

The proof closely mirrors the proof of the nonconstructive upper bound, Theorem 1, but using the constructive constants $C_\ell, H_L, \epsilon_\psi$ derived above. To summarize, we have shown the following.

**Theorem 6.** *Let $\ell : \mathcal{R} \to \mathbb{R}_+^{\mathcal{Y}}$ be a discrete target loss, $L : \mathbb{R}^d \to \mathbb{R}_+^{\mathcal{Y}}$ be a polyhedral surrogate loss, and $\psi : \mathbb{R}^d \to \mathcal{R}$ a link function. The following are equivalent:*

1. *$(L, \psi)$ is consistent for $\ell$.*

2. *$\psi$ is $\epsilon$-separated with respect to $L$ and $\ell$ for some $\epsilon > 0$.*

3. *$(L, \psi)$ guarantees a linear regret transfer for $\ell$.*

*Furthermore, if any of the above hold, then in particular $(L, \psi)$ guarantees the regret transfer $\zeta(t) = \left( \frac{C_\ell H_L}{\epsilon_\psi} \right) t$.*

## 5.2 Further tightening

While our goal is not to focus on exact constants for specific problems, we remark on a few ways Theorem 5 may be loose and how one could compute the tightest possible constant $\alpha^*$ for which $R_\ell(\psi \circ h; \mathcal{D}) \leq \alpha^* R_L(h; \mathcal{D})$ for all $h$ and $\mathcal{D}$. In general, for a fixed $p$, there is some smallest $\alpha_p^*$ such that $R_\ell(\psi(u), p) \leq \alpha_p^* R_L(u, p)$ for all $u$. Then, it follows from our results that $\alpha^* = \max_{p \in \mathcal{Q}} \alpha_p^*$ for the finite set $\mathcal{Q}$ used in the proof, i.e. the vertices of the full-dimensional level sets of $\Gamma = \mathrm{prop}[L]$.

Above, we bounded $\alpha_p^* \leq \frac{C_\ell H_{L,p}}{\epsilon_\psi}$. The intuition is that some $u$ at distance $\geq \epsilon_\psi$ from $\Gamma(p)$, the optimal set, may link to a "bad" report $r = \psi(u) \notin \gamma(p)$. The rate at which $L$ grows is at least $H_{L,p}$, so the surrogate loss at $u$ may be as small as $\frac{\epsilon_\psi}{H_{L,p}}$, while the target regret may be as high as $C_\ell = \max_{r', p'} R_\ell(r', p')$. The ratio of regrets is therefore bounded by $\frac{H_{L,p} C_\ell}{\epsilon_\psi}$.

The tightest possible bound, on the other hand, is $\alpha^* = \sup_{u : \psi(u) \notin \gamma(p)} \frac{R_\ell(\psi(u), p)}{R_L(u, p)}$. This bound can be smaller if the values of numerator and denominator are correlated across $u$. For example, $u$ may only be $\epsilon_\psi$-close to the optimal set when it links to reports $\psi(u)$ with lower target regret; or $L$ may have a smaller slope in the direction where the link's separation is larger than $\epsilon$.

To illustrate with a concrete example, consider the *binary encoded predictions (BEP) surrogate* of [25] for the abstain target loss, $\ell(r, y) = \frac{1}{2}$ if $r = \perp$, otherwise $\ell(r, y) = \mathbb{1}[r \neq y]$. The surrogate involves an injective map $B : \mathcal{Y} \to \{-1, 1\}^d$ for $d = \lceil \log_2 |\mathcal{Y}| \rceil$. It is $L(u)_y = \max_{j=1 \ldots d} (1 - u_d B(y)_d)_+$, where $(\cdot)_+$ indicates taking the maximum of the argument and zero. The associated link is $\psi(u) = \perp$ if $\min_{j=1 \ldots d} |u_j| \leq \frac{1}{2}$, otherwise $\psi(u) = \arg \min_{y \in \mathcal{Y}} \|B(y) - u\|_\infty$.

One can show that for $p = \delta_y$, i.e. the distribution with full support on some $y \in \mathcal{Y}$, $L(u)_y = d_\infty(u, \Gamma(p))$ exactly, giving $H_{L,p} = 1$. It is almost immediate that $\epsilon_\psi = \frac{1}{2}$. Meanwhile, $R_\ell(r, p) \leq 1$, giving us an upper bound $\alpha_p^* \leq \frac{(1)(1)}{1/2} = 2$. In fact, this is slightly loose; the exact constant, given in [25] is 1. The looseness stems from the fact that for $p = \delta_y$, the closest reports $u$ to the optimal set, i.e. at distance only $\epsilon_\psi = \frac{1}{2}$ away, do not link to reports maximizing target regret; they link to the abstain report $\perp$, which has regret only $\frac{1}{2}$. With this correction, and an observation that all $u$ linking to reports $y' \neq y$ are at distance at least $\frac{3}{2}$ from $\Gamma(p)$, we restore the tight bound $\alpha_p^* \leq 1$. A similar but slightly more involved calculation can be carried out for the other vertices $p \in \mathcal{Q}$, which turn out to be all vertices of the form $\frac{1}{2} \delta_y + \frac{1}{2} \delta_{y'}$.

Finally, while we use $\|\cdot\|_\infty$ to define the minimum-slope $H_L$ and the separation $\epsilon_\psi$, in principle one could use another norm. One reason for restricting to $\|\cdot\|_\infty$ is that it is more compatible with Hoffman constants. However, all definitions hold for other norms and so does the main upper bound, as existence of an $H_L$ and $\epsilon_\psi$ in $\|\cdot\|_\infty$ imply existence of constants for other norms. These constants may change for different norms, and in particular, the optimal overall constant may arise from a norm other than $\|\cdot\|_\infty$.

# 6 Discussion

We have shown two broad results about regret tranfer functions for surrogate losses. In particular, polyhedral surrogates always achieve a linear transfer, whereas "non-polyhedral" surrogates are generally square root. Section 5 outlines several directions to further refine the bound for polyhedral surrogates. Beyond these directions, an interesting question is to what extent our results hold when the label set or report set are infinite. For example, when the labels are the real line, pinball loss is polyhedral yet elicits a quantile, a very different case than the one we study. In particular, Lemma 4 will not give a finite set of optimal sets in this case. Nonetheless, we suspect similar results could go through under some regularity assumptions. Finally, in line with our results, we would like to better understand the tradeoff between smoothness of the surrogate and the overall generalization rate. One important direction along these lines is to extend the analysis of Mahdavi et al. [17] beyond binary classification.

## Broader Impacts

As a theoretical work, the likely impacts of this paper take the form of downstream research and applications. Instead of direct applications, we anticipate this work leading to more investigation of surrogate losses to improve discrete prediction tasks. It may inform practitioners' choices of which surrogate losses they use for supervised learning tasks. Of course, such machine learning tasks can be solved for ethical or unethical purposes. We do not know of particular risk of negative impacts of this work beyond risks of supervised machine learning in general.

## Acknowledgments and Disclosure of Funding

We thank Stephen Becker, Jessie Finocchiaro, and Nishant Mehta for insights, discussions, and references. This material is based upon work supported by the National Science Foundation under Grant No. IIS-2045347.

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
