# A Omitted Proofs: Upper Bound

This section contains omitted proofs from Section 3.

**Lemma** (Lemma 1). *Let $\ell : \mathcal{R} \to \mathbb{R}_+^{\mathcal{Y}}$ be a discrete target loss and suppose the surrogate $L : \mathbb{R}^d \to \mathbb{R}_+^{\mathcal{Y}}$ and link $\psi : \mathbb{R}^d \to \mathcal{R}$ are calibrated for $\ell$. Then for any $p \in \Delta_{\mathcal{Y}}$, there exists $\alpha_p \geq 0$ such that, for all $u \in \mathbb{R}^d$,*

$$R_\ell(\psi(u), p) \leq \alpha_p R_L(u, p).$$

*Proof.* Fix $p \in \Delta_{\mathcal{Y}}$. Let $C_p = \max_{r \in \mathcal{R}} R_\ell(r, p)$. The maximum exists because $\ell$ is discrete, i.e. $\mathcal{R}$ is finite. Meanwhile, recall that, when defining calibration, we let $B_{L,\psi,\ell}(p) = \{R_L(u, p) : \psi(u) \notin \gamma(p)\}$. Let $B_p = \inf B_{L,\psi,\ell}(p)$. By definition of calibration, we have $B_p > 0$.

To combine these bounds, let $\alpha_p = \frac{C_p}{B_p}$. Let $u \in \mathbb{R}^d$. There are two cases. If $\psi(u) \in \gamma(p)$, then $R_\ell(\psi(u), p) = 0 \leq R_L(u, p)$ immediately. If $\psi(u) \notin \gamma(p)$, then

$$\begin{aligned} R_\ell(\psi(u), p) &\leq C_p \\ &= \alpha_p \cdot B_p \\ &\leq \alpha_p R_L(u, p). \end{aligned}$$

$\square$

**Lemma** (Lemma 2). *If $(L, \psi)$ indirectly elicits $\ell$, then $\Gamma = \mathrm{prop}[L]$ refines $\gamma = \mathrm{prop}[\ell]$ in the sense that, for all $u \in \mathbb{R}^d$, there exists $r \in \mathcal{R}$ such that $\Gamma_u \subseteq \gamma_r$.*

*Proof.* For any $u$, let $r = \psi(u)$. By indirect elicitation, $u \in \Gamma(p) \implies r \in \gamma(p)$. So $\Gamma_u = \{p : u \in \Gamma(p)\} \subseteq \{p : r \in \gamma(p)\} = \gamma_r$. $\square$

**Lemma** (Lemma 3). *Suppose $(L, \psi)$ indirectly elicits $\ell$ and let $\Gamma = \mathrm{prop}[L]$. Then for any fixed $u, u^* \in \mathbb{R}^d$ and $r \in \mathcal{R}$, the functions $R_L(u, \cdot)$ and $R_\ell(r, \cdot)$ are linear in their second arguments on $\Gamma_{u^*}$.*

*Proof.* Let $u^* \in \mathbb{R}^d$ and $p \in \Gamma_{u^*}$. By definition, for all $p \in \Gamma_{u^*}$, $\underline{L}(p) = \langle p, L(u^*) \rangle$. So for fixed $u$,

$$R_L(u, p) = \langle p, L(u) \rangle - \langle p, L(u^*) \rangle = \langle p, L(u) - L(u^*) \rangle,$$

a linear function of $p$ on $\Gamma_{u^*}$. Next, by Lemma 2, there exists $r^*$ such that $\Gamma_{u^*} \subseteq \gamma_{r^*}$. By the same argument, for fixed $r$, $R_\ell(r, p) = \langle p, \ell(r) - \ell(r^*) \rangle$, a linear function of $p$ on $\gamma_{r^*}$ and thus on $\Gamma_{u^*}$. $\square$

**Lemma** (Lemma 4). *If $L : \mathbb{R}^d \to \mathbb{R}_+^{\mathcal{Y}}$ is polyhedral, then $\Gamma = \mathrm{prop}[L]$ has a finite set of level sets that union to $\Delta_{\mathcal{Y}}$. Moreover, these level sets are polytopes.*

*Proof.* This statement can be deduced from the embedding framework of [10]. In particular, Lemma 5 of [10] states that if $L$ is polyhedral, then its Bayes risk $\underline{L}$ is concave polyhedral, i.e. is the pointwise minimum of a finite set of affine functions. It follows that there exists a finite set $U \subset \mathbb{R}^d$ such that

$$\underline{L}(p) = \min_{u \in \mathbb{R}^d} \langle p, L(u) \rangle = \min_{u \in U} \langle p, L(u) \rangle . \tag{4}$$

We claim the level sets of $U$ witness the claim. First, it is known (e.g. from theory of power diagrams, [2]) that if $\underline{L}$ is a polyhedral function represented as (4) and $u \in U$, then $\Gamma_u = \{p \in \Delta_{\mathcal{Y}} : \langle p, L(u) \rangle = \underline{L}(p)\}$ is a polytope. Finally, suppose for contradiction that there exists $p \in \Delta_{\mathcal{Y}}$, $p \notin \cup_{u \in U} \Gamma_u$. Then there must be some $u' \notin U$ with $p \in \Gamma_{u'}$, implying that $\langle p, L(u') \rangle > \max_{u \in U} \langle p, L(u) \rangle$, contradicting (4). $\square$

**Theorem** (Theorem 3). *Suppose the surrogate loss $L : \mathbb{R}^d \to \mathbb{R}_+^{\mathcal{Y}}$ and link $\psi : \mathbb{R}^d \to \mathcal{R}$ are consistent for the target loss $\ell : \mathcal{R} \to \mathbb{R}_+^{\mathcal{Y}}$. If $L$ is polyhedral, then $(L, \psi)$ guarantee a linear regret transfer for $\ell$, i.e. there exists $\alpha \geq 0$ such that, for all $\mathcal{D}$ and all measurable $h : \mathcal{X} \to \mathbb{R}^d$,*

$$R_\ell(\psi \circ h; \mathcal{D}) \leq \alpha R_L(h; \mathcal{D}).$$

*Proof.* We first recall that by Fact 1, consistency implies that $(L, \psi)$ are calibrated for $\ell$ and that $(L, \psi)$ indirectly elicit $\ell$. Next, by Observation 1, it suffices to show a linear *conditional* regret transfer, i.e. for all $p \in \Delta_{\mathcal{Y}}$ and $u \in \mathbb{R}^d$, we show $R_\ell(\psi(u), p) \leq \alpha R_L(u, p)$.

By Lemma 4, the polyhedral loss $L$ has a finite set $U \subset \mathbb{R}^d$ of predictions such that (a) for each $u \in U$, the level set $\Gamma_u$ is a polytope, and (b) $\cup_{u \in U} \Gamma_u = \Delta_{\mathcal{Y}}$. Let $\mathcal{Q}_u \subset \Delta_{\mathcal{Y}}$ be the finite set of vertices of the polytope $\Gamma_u$, and define the finite set $\mathcal{Q} = \cup_{u \in U} \mathcal{Q}_u$.

By Lemma 1, for each $q \in \mathcal{Q}$, there exists $\alpha_q \geq 0$ such that $R_\ell(\psi(u), q) \leq \alpha_q R_L(u, q)$ for all $u$. We choose

$$\alpha = \max_{q \in \mathcal{Q}} \alpha_q.$$

To prove the conditional regret transfer, consider any $p \in \Delta_{\mathcal{Y}}$ and any $u \in \mathbb{R}^d$. There exists $u \in U$ such that $p \in \Gamma_u$, a polytope. So we can write $p$ as a convex combination of its vertices, i.e.

$$p = \sum_{q \in \mathcal{Q}_u} \beta(q) q$$

for some probability distribution $\beta$. Recall that $\mathcal{Q}_u \subseteq \Gamma_u$ and $R_L$ and $R_\ell$ are linear in $p$ on $\Gamma_u$ by Lemma 3. So, for any $u'$:

$$
\begin{aligned}
R_\ell(\psi(u'), p) &= R_\ell \left( \psi(u'), \sum_{q \in \mathcal{Q}_u} \beta(q) q \right) \\
&= \sum_{q \in \mathcal{Q}_u} \beta(q) R_\ell(\psi(u'), q) \\
&\leq \sum_{q \in \mathcal{Q}_u} \beta(q) \alpha_q R_L(u', q) \\
&\leq \alpha \sum_{q \in \mathcal{Q}_u} \beta(q) R_L(u', q) \\
&= \alpha R_L(u', p).
\end{aligned}
$$

$\square$

## B    Omitted Proofs: Lower Bound

This section contains omitted proofs from Section 4.

**Theorem** (Theorem 4). *Suppose the surrogate loss $L$ and link $\psi$ satisfy a regret transfer of $\zeta$ for a target loss $\ell$. If $L$, $\psi$, and $\ell$ satisfy Assumption 1, then there exists $c > 0$ such that, for some $\epsilon^* > 0$, for all $0 \leq \epsilon < \epsilon^*$, $\zeta(\epsilon) \geq c\sqrt{\epsilon}$.*

*Proof outline:* By assumption we have a boundary report $u_0$ which is $L$-optimal for a distribution $p_0$. We have some $r, r'$ which are both optimal for $p_0$, and $\psi(u_0) = r'$. First, we will choose a $p_1$ where $r$ is uniquely optimal, hence $u_0$ is a strictly suboptimal choice. We then consider a sequence of distributions $p_\lambda = (1 - \lambda)p_0 + \lambda p_1$, approaching $p_0$ as $\lambda \to 0$. For all such $p_\lambda$, it will happen that $r$ is optimal while $u_0$ and $r' = \psi(u_0)$ are strictly suboptimal. We show that $R_\ell(r', p_\lambda) = c_\ell \lambda$ for some constant $c_\ell$ and all small enough $\lambda$. Meanwhile, we will show that $R_L(u_0, p_\lambda) \leq O(\lambda^2)$, proving the result. The last fact will use the properties of strong smoothness and strong convexity in a neighborhood of $u_0$.

*Proof.* Obtain $\alpha, u_0, p_0, r, r'$, and an open neighborhood of $u_0$ from Assumption 1 and the definition of boundary report. Assume without loss of generality that $\psi(u_0) = r'$; otherwise, swap the roles of $r$ and $r'$.

**Linearity of $R_\ell(r', p_\lambda)$.** As $\ell$ is non-redundant by assumption, there exists some $p_1 \in \mathring{\gamma}_r$, the relative interior of the full-dimensional level set $\gamma_r$. We therefore have $R_\ell(r', p_1) = \langle p_1, \ell(r') - \ell(r) \rangle =: c_\ell > 0$, and $R_\ell(r', p_0) = 0$. Let $p_\lambda := (1 - \lambda)p_0 + \lambda p_1$. By convexity of $\gamma_r$, we have $p_\lambda \in \gamma_r$ for all $\lambda \in [0, 1]$, which gives $R_\ell(r', p_\lambda) = \lambda c_\ell$.

**Obtaining the global minimizer $u_\lambda$ of $L_\lambda$.** Let $L_\lambda : \mathbb{R}^d \to \mathbb{R}_+$ be given by $L_\lambda(u) = \langle p_\lambda, L(u) \rangle = (1 - \lambda)\langle p_0, L(u) \rangle + \lambda \langle p_1, L(u) \rangle$. Let $\delta > 0$ such that the above open neighborhood of $u_0$ contains the Euclidean ball $B_\delta(u_0)$ of radius $\delta$ around $u_0$. Let $u_1 \in \Gamma(p_1)$. We argue that for all small enough $\lambda$, $L_\lambda(u)$ is uniquely minimized by some $u_\lambda \in B_\delta(u_0)$. For any $u \notin B_\delta(u_0)$, we have, using local strong convexity and the optimality of $u_1$,

$$
\begin{aligned}
L_\lambda(u) - L_\lambda(u_0) &= (1 - \lambda)\left(L_0(u) - L_0(u_0)\right) + \lambda\left(L_1(u) - L_1(u_0)\right) \\
&\geq (1 - \lambda)\left(\frac{\alpha}{2}\delta^2\right) + \lambda\left(L_1(u_1) - L_1(u_0)\right) \\
&> 0
\end{aligned}
$$

if $\lambda < \lambda^* := \alpha\delta^2/(2\alpha\delta^2 + 4L_1(u_0) - 4L_1(u_1))$. For the remainder of the proof, let $\lambda < \lambda^*$. Then any $u \notin B_\delta(u_0)$ has $L_\lambda(u) > L_\lambda(u_0)$, hence is suboptimal. By $\alpha$-strong convexity of $L_0$ on $B_\delta(u_0)$, $L_\lambda$ is strictly convex on $B_\delta(u_0)$. So it has a unique minimizer $u_\lambda$, and by the above argument this is the global minimizer of $L_\lambda$. Then $\underline{L}(p_\lambda) = L_\lambda(u_\lambda)$, and thus $R_L(u_0, p_\lambda) = L_\lambda(u_0) - L_\lambda(u_\lambda)$. We also observe here that $R_L(u_0, p_\lambda)$ is continuous in $\lambda$, e.g. because the Bayes risk of $L$ is continuous in $p$ as is $\langle p, L(u_0) \rangle$. It is also zero when $\lambda = 0$.

**Showing $R_L$ is quadratic in $\lambda$.** By assumption, the gradient of $L_y$ is locally Lipschitz for all $y \in \mathcal{Y}$. We will apply this fact to the compact set $\mathcal{C} = \{u \in \mathbb{R}^d : \|u - u_1\| \leq \|u_0 - u_1\| + \delta\}$. By compactness, we have a finite subcover of open neighborhoods; let $\beta$ be the minimum Lipschitz constant over this finite set of neighborhoods. We thus have that $L_y$ is $\beta$-strongly smooth on $\mathcal{C}$, and hence so is $L_\lambda$ for any $\lambda \in [0, 1]$.

We now upper bound $\|u_\lambda - u_0\|_2$, and then apply strong smoothness to upper bound $R_L(u_0, p_\lambda) = L_\lambda(u_0) - L_\lambda(u_\lambda)$. Consider the first-order optimality condition of $L_\lambda$:

$$
\begin{aligned}
0 = \nabla L_\lambda(u_\lambda) &= (1 - \lambda)\nabla L_0(u_\lambda) + \lambda \nabla L_1(u_\lambda) \\
&\implies (1 - \lambda)\|\nabla L_0(u_\lambda)\|_2 = \lambda\|\nabla L_1(u_\lambda)\|_2 .
\end{aligned}
$$

By optimality of $u_0$ and $u_1$, strong convexity of $L_0$ and strong smoothness of $L_1$, and the triangle inequality, we have

$$
\begin{aligned}
\|\nabla L_0(u_\lambda)\|_2 &= \|\nabla L_0(u_\lambda) - \nabla L_0(u_0)\|_2 \geq \alpha\|u_\lambda - u_0\|_2 , \\
\|\nabla L_1(u_\lambda)\|_2 &= \|\nabla L_1(u_\lambda) - \nabla L_1(u_1)\|_2 \leq \beta\|u_\lambda - u_1\|_2 \\
&\leq \beta\left(\|u_\lambda - u_0\|_2 + \|u_0 - u_1\|_2\right) .
\end{aligned}
$$

Combining,

$$
\begin{aligned}
(1 - \lambda)\alpha\|u_\lambda - u_0\|_2 &\leq (1 - \lambda)\|\nabla L_0(u_\lambda)\|_2 \\
&= \lambda\|\nabla L_1(u_\lambda)\|_2 \\
&\leq \lambda\beta\left(\|u_\lambda - u_0\|_2 + \|u_0 - u_1\|_2\right) .
\end{aligned}
$$

Now rearranging and taking $\lambda \leq \frac{1}{2}\frac{\alpha}{\alpha+\beta}$, we have

$$
\|u_\lambda - u_0\|_2 \leq \frac{\lambda\beta}{(1 - \lambda)\alpha - \lambda\beta}\|u_0 - u_1\|_2 \leq \lambda\frac{2\beta}{\alpha}\|u_0 - u_1\|_2 .
$$

Finally, from strong smoothness of $L_\lambda$ and optimality of $u_\lambda$,

$$
L_\lambda(u_0) - L_\lambda(u_\lambda) \leq \frac{\beta}{2}\|u_0 - u_\lambda\|_2^2 \leq \frac{\beta}{2}\left(\lambda\frac{2\beta}{\alpha}\|u_0 - u_1\|_2\right)^2 = c_L\lambda^2 ,
$$

where $c_L = \frac{2\beta^3}{\alpha^2}\|u_0 - u_1\|_2^2 > 0$.

To conclude: we have found a $\lambda^* > 0$ and shown that for all $0 \leq \lambda < \lambda^*$, $R_\ell(r', p_\lambda) = c_\ell\lambda$ and $R_L(u_0, p_\lambda) \leq c_L\lambda^2$. In particular, let $\epsilon^* = \sup_{0 \leq \lambda < \lambda^*} R_L(u_0, p_\lambda)$. Then for all $0 \leq \epsilon < \epsilon^*$, by continuity, we can choose $\lambda < \lambda^*$ such that $R_L(u_0, p_\lambda) = \epsilon \leq c_L\lambda^2$. Meanwhile, $R_\ell(\psi(u_0), p_\lambda) = c_\ell\lambda \geq \frac{c_\ell}{\sqrt{c_L}}\sqrt{\epsilon}$. Recalling that $\zeta(R_L(u_0, p_\lambda)) \geq R_\ell(\psi(u_0), p_\lambda)$ by definition, this implies $\zeta(\epsilon) \geq c\sqrt{\epsilon}$ for all $\epsilon < \epsilon^*$, with $c = \frac{c_\ell}{\sqrt{c_L}}$. $\qquad\square$

# C  Omitted Proofs: Constant Derivation

This section contains omitted proofs from Section 5.

## C.1  Hoffman constants

First we appeal to a known fact, the existence of Hoffman constants for systems of linear inequalities. See Zalinescu [35] for a modern treatment.

**Theorem 7** (Hoffman constant [14]). *Given a matrix $A \in \mathbb{R}^{m \times n}$, there exists some smallest $H(A) \geq 0$, called the* Hoffman constant *(with respect to $\| \cdot \|_\infty$), such that for all $b \in \mathbb{R}^m$ and all $x \in \mathbb{R}^n$,*

$$d_\infty(x, S(A, b)) \leq H(A)\|(Ax - b)_+\|_\infty , \tag{5}$$

*where $S(A, b) = \{x \in \mathbb{R}^n \mid Ax \leq b\}$ and $(u)_+ \doteq \max(u, 0)$ component-wise.*

**Lemma** (Lemma 5). *Let $L : \mathbb{R}^d \to \mathbb{R}^{\mathcal{Y}}_+$ be a polyhedral loss with $\Gamma = \mathrm{prop}[L]$. Then for any fixed $p$, there exists some smallest constant $H_{L,p} \geq 0$ such that $d_\infty(u, \Gamma(p)) \leq H_{L,p} R_L(u, p)$ for all $u \in \mathbb{R}^d$.*

*Proof.* Since $L$ is polyhedral, there exist $a_1, \ldots, a_m \in \mathbb{R}^d$ and $c \in \mathbb{R}^m$ such that we may write $\langle p, L(u) \rangle = \max_{1 \leq j \leq m} a_j \cdot u + c_j$. Let $A \in \mathbb{R}^{m \times d}$ be the matrix with rows $a_j$, and let $b = \underline{L}(p)\mathbb{1} - c$, where $\mathbb{1} \in \mathbb{R}^m$ is the all-ones vector. Then we have

$$\begin{aligned}
S(A, b) &\doteq \{u \in \mathbb{R}^d \mid Au \leq b\} \\
&= \{u \in \mathbb{R}^d \mid Au + c \leq \underline{L}(p)\mathbb{1}\} \\
&= \{u \in \mathbb{R}^d \mid \forall i \, (Au + c)_i \leq \underline{L}(p)\} \\
&= \{u \in \mathbb{R}^d \mid \max_i \, (Au + c)_i \leq \underline{L}(p)\} \\
&= \{u \in \mathbb{R}^d \mid \langle p, L(u) \rangle \leq \underline{L}(p)\} \\
&= \Gamma(p) .
\end{aligned}$$

Similarly, we have $\max_i \, (Au - b)_i = \langle p, L(u) \rangle - \underline{L}(p) = R_L(u, p) \geq 0$. Thus,

$$\begin{aligned}
\|(Au - b)_+\|_\infty &= \max_i \, ((Au - b)_+)_i \\
&= \max((Au - b)_1, \ldots, (Au - b)_m, 0) \\
&= \max(\max_i \, (Au - b)_i, \, 0) \\
&= \max_i \, (Au - b)_i \\
&= R_L(u, p) .
\end{aligned}$$

Now applying Theorem 7, we have

$$\begin{aligned}
d_\infty(u, \Gamma(p)) &= d_\infty(u, S(A, b)) \\
&\leq H(A)\|(Au - b)_+\|_\infty \\
&= H(A) R_L(u, p) .
\end{aligned}$$

$\square$

## C.2  Separated links

**Lemma** (Lemma 6). *Let polyhedral surrogate $L : \mathbb{R}^d \to \mathbb{R}^{\mathcal{Y}}_+$, discrete loss $\ell : \mathcal{R} \to \mathbb{R}^{\mathcal{Y}}_+$, and link $\psi : \mathbb{R}^d \to \mathcal{R}$ be given such that $(L, \psi)$ is calibrated with respect to $\ell$. Then there exists $\epsilon > 0$ such that $\psi$ is $\epsilon$-separated with respect to $\Gamma \doteq \mathrm{prop}[L]$ and $\gamma \doteq \mathrm{prop}[\ell]$.*

*Proof.* Suppose that $\psi$ is not $\epsilon$-separated for any $\epsilon > 0$. Then letting $\epsilon_i \doteq 1/i$ we have sequences $\{p_i\}_i \subset \Delta_{\mathcal{Y}}$ and $\{u_i\}_i \subset \mathbb{R}^d$ such that for all $i \in \mathbb{N}$ we have both $\psi(u_i) \notin \gamma(p_i)$ and $d_\infty(u_i, \Gamma(p_i)) \leq \epsilon_i$. First, observe that there are only finitely many values for $\gamma(p_i)$ and $\Gamma(p_i)$,

as $\mathcal{R}$ is finite and $L$ is polyhedral. Thus, there must be some $p \in \Delta_{\mathcal{Y}}$ and some infinite subsequence indexed by $j \in J \subseteq \mathbb{N}$ where for all $j \in J$, we have $\psi(u_j) \notin \gamma(p)$ and $\Gamma(p_j) = \Gamma(p)$.

Next, observe that, as $L$ is polyhedral, the expected loss $\langle p, L(u) \rangle$ is $\beta$-Lipschitz in $\| \cdot \|_\infty$ for some $\beta > 0$. Thus, for all $j \in J$, we have

$$
\begin{aligned}
d_\infty(u_i, \Gamma(p)) \leq \epsilon_j &\implies \exists u^* \in \Gamma(p) \| u_j - u^* \|_\infty \leq \epsilon_j \\
&\implies |\langle p, L(u_j) \rangle - \langle p, L(u^*) \rangle| \leq \beta \epsilon_j \\
&\implies |\langle p, L(u_j) \rangle - \underline{L}(p)| \leq \beta \epsilon_j \ .
\end{aligned}
$$

Finally, for this $p$, we have

$$
\inf_{u : \psi(u) \notin \gamma(p)} \langle p, L(u) \rangle \leq \inf_{j \in J} \langle p, L(u_j) \rangle = \underline{L}(p) \ ,
$$

contradicting the calibration of $\psi$. $\qquad \square$

## C.3 Combining the loss and link

**Lemma 7.** *Let $\ell : \mathcal{R} \to \mathbb{R}_+^{\mathcal{Y}}$ be a discrete target loss, $L : \mathbb{R}^d \to \mathbb{R}_+^{\mathcal{Y}}$ be a polyhedral surrogate loss, and $\psi : \mathbb{R}^d \to \mathcal{R}$ a link function. If $(L, \psi)$ indirectly elicit $\ell$ and $\psi$ is $\epsilon$-separated, then for all $u$ and $p$,*

$$
R_\ell(\psi(u), p) \leq \frac{C_\ell H_{L,p}}{\epsilon} R_L(u, p).
$$

*Proof.* If $\psi(u) \in \gamma(p)$, then $R_\ell(u, p) = 0$ and we are done. Otherwise, applying the definition of $\epsilon$-separated and Lemma 5,

$$
\begin{aligned}
\epsilon &< d_\infty(u, \Gamma(p)) \\
&\leq H_{L,p} R_L(u, p).
\end{aligned}
$$

So $R_\ell(\psi(u), p) \leq C_\ell \leq \frac{C_\ell H_{L,p}}{\epsilon} R_L(u, p)$. $\qquad \square$

**Theorem** (Constructive linear transfer, Theorem 5). *Let $\ell : \mathcal{R} \to \mathbb{R}_+^{\mathcal{Y}}$ be a discrete target loss, $L : \mathbb{R}^d \to \mathbb{R}_+^{\mathcal{Y}}$ be a polyhedral surrogate loss, and $\psi : \mathbb{R}^d \to \mathcal{R}$ a link function. If $(L, \psi)$ are consistent for $\ell$, then*

$$
(\forall h, \mathcal{D}) \quad R_\ell(\psi \circ h; \mathcal{D}) \leq \frac{C_\ell H_L}{\epsilon_\psi} R_L(h; \mathcal{D}) \ .
$$

The proof closely mirrors the proof of the nonconstructive upper bound, Theorem 1.

*Proof.* By Lemma 6, $\psi$ is separated and $\epsilon_\psi$ well-defined. By Lemma 7, for each $p \in \mathcal{Q}$, $R_\ell(\psi(u), p) \leq \frac{C_\ell H_L}{\epsilon_\psi} R_L(u, p)$ for all $u$. Now consider a general $p$, which is in some full-dimensional polytope level set $\Gamma_u$. Write $p = \sum_{q \in \mathcal{Q}_u} \beta(q) q$ for some probability distribution $\beta$, where $\mathcal{Q}_u$ is the set of vertices of $\Gamma_u$. By Lemma 3, $R_L$ and $R_\ell$ are linear in $p$ on $\Gamma_u$, so for any $u'$,

$$
\begin{aligned}
R_\ell(\psi(u'), p) &= \sum_{q \in \mathcal{Q}_u} \beta(q) R_\ell(\psi(u'), q) \\
&\leq \sum_{q \in \mathcal{Q}_u} \beta(q) \frac{C_\ell H_{L,p}}{\epsilon_\psi} R_L(u', q) \\
&\leq \frac{C_\ell H_L}{\epsilon_\psi} \sum_{q \in \mathcal{Q}_u} \beta(q) R_L(u', q) \\
&\leq \frac{C_\ell H_L}{\epsilon_\psi} R_L(u', p).
\end{aligned}
$$

By Observation 1, this conditional regret transfer implies a full regret transfer, with the same constant. $\qquad \square$