# OpenReview forum: "Surrogate Regret Bounds for Polyhedral Losses"
_NeurIPS.cc/2021/Conference — NeurIPS 2021 Poster_

### Official Review · Reviewer_auqc · 2021-07-14

**Rating:** 8
**Confidence:** 3

**Summary:**

This paper provides general excess risk bounds result for a class of polyhedral losses for a general machine learning problem, where a prediction space and label space lie in a finite set. This covers many fundamental problems, e.g., classification, ranking, classification with rejection.  While such theoretical results have been quite well-studied for a family of strictly proper losses (a loss that can estimate the class probability (p(y|x)), less attention has been paid to other loss families. This paper shows that a polyhedral surrogate loss (i.e., piecewise-linear convex loss such as hinge loss) that is consistent with the target discrete loss has a linear excess risk transfer bound, which is desirable. Furthermore, the paper showed that a large class of non-polyhedral loss (including some strictly proper losses) cannot achieve the same result. On the other hand, such non-polyhedral losses have a slower rate than linear (square root).

**Ethical Concerns:**

The discussion this paper provided is adequate.

**Limitations And Societal Impact:**

The discussion this paper provided is adequate.

**Main Review:**

The main contributions of this paper are clear, impactful, and well-presented. It completely characterizes the excess risk transfer bound of any convex polyhedral losses. The discussion of related work is also appropriate and nice. The constructive linear transfer kindly provided in Section 5 is impressive. On the other hand, it would have been better if there are some illustrations (e.g., Figure, table) to help readers to understand this paper. For example, an illustration of the shape of polyhedral losses, a table to highlight the well-known non-polyhedral loss functions that Theorem 2 covers and does not cover.

Questions and comments:
1. This paper delivers a strong theoretical result in the case where a prediction space and label space are finite sets. My impression is that this result is already very general in this submission. Is there a case where we have non-finite sets for prediction/label space yet polyhedral losses are used in the literature up until now, where the theory of this paper does not cover?
2. Line 55: It seems the previous result by Finocchiaro et al. [8] suggested that the Lovasz hinge loss, in general, is inconsistent, and perhaps it is not applicable to Theorem 1 and the excess risk transfer bound should not exist. Maybe a few words about this in the manuscript would be useful if my understanding is correct.

Minor comments:
1. Line 47: suprisingly -> surprisingly
2. Line 109: Theeeorem - > Theorem
3. Ref [21] in the manuscript: In my understanding, this work was only done by Clayton Scott (single author) and therefore "et al." should be removed
4. Ref [28] in the manuscript: In my understanding, this work was only done by Tong Zhang (single author) and therefore "et al." should be removed

More references related to calibration/excess risk bound for other specific problems that the authors may want to include in the manuscript.

Binary classification with rejection:

[1] Cortes et al., Learning with rejection, ALT, 2016 (Theorem 3 - a variant of hinge loss with linear excess risk bound).
FYI: There exists another work by the same authors where they used an exponential-based loss for this problem with calibration guarantee [2], but unfortunately the excess risk bound was not provided to my knowledge (but I guess it would be slower than linear).

[2] Cortes et al., Boosting with abstention, NeurIPS, 2016

Multiclass classification with rejection:

[3] Ni et al., On the calibration of multiclass classification with rejection, NeurIPS, 2019 (Theorem 7 (ฺBound for OVA loss based on strictly proper composite loss choice (square root rate for many losses according to Table 1)) and Theorem 8 (Bound for softmax cross-entropy loss (square root rate)).

[4] Charoenphakdee et al., Classification with rejection based on cost-sensitive classification, ICML, 2021 (Theorem 8 (excess risk transfer bound for classification with rejection and general classification-calibrated loss) and Corollary 9 (Excess risk bound of hinge loss with linear rate)).

Bipartite ranking:

[5] Gao and Zhou, On the consistency of AUC pairwise optimization, IJCAI, 2015 (Theorem 3 and Corollary 5, where Corollary 5 indicates that exponential and logistic losses give a square root regret bound).

----- Updated comment after receiving the author feedback-----

I have read other reviews and the author feedback. I appreciate the authors for the response.

I found that the contributions of this paper are clear and useful as a general theory of polyhedral loss for any discrete target, that is, for any discrete target, there exists a polyhedral loss that has a linear regret transfer rate for it. In addition, this paper discusses a large class of loss functions (that is non-polyhedral but not exactly all possible non-polyhedral losses) that can always achieve a rate slower than linear. Both results are novel to the best of my knowledge and I think it is safe to say that this provides a solid understanding of polyhedral losses. While there exist several existing works that proved the regret transfer rate of a loss function (both polyhedral and non-polyhedral), it is usually done for a specific loss function, where the proof usually uses the specific property of that loss and it cannot be used for other losses. With the results of this submission, we can immediately use it for confirming the rate of convergence of a new loss function as long as it satisfies the condition (i.e., polyhedral or a class of non-polyhedral losses defined in this paper). Thus, it is quite clear that this paper is novel and useful and this is the main reason why I vote to accept this paper.

On weakness pointed out by other reviewers:

As we have seen in several existing works that polyhedral losses often (or always?) achieve the linear rate while it is not the case for many smooth losses, the result in this paper may not be very surprising. Nevertheless, I believe it is important to have a confirmation that this is indeed true in general. And it is nice to know that we can actually prove a general statement as suggested by the authors. Another weakness of this paper could be that there is no experiment. But I think this paper focuses on the theoretical perspective and it already has sufficient contribution for NeurIPS. The extensive empirical study can be done later as future work. Therefore, my score remains unchanged.




**Time Spent Reviewing:**

10 hours

---

> ### Author Response · Authors · 2021-08-10
> **Response to questions and comments**
>
> 1. Thank you for this great question, which we should comment on in the paper. Indeed, pinball loss, used in quantile regression, is such an example: pinball loss is polyhedral but the label space is the real line. (A simple special case is absolute loss for median regression.) Our results do not apply in this case immediately, since Lemma 4 (establishing that there are a finite number of possible optimal sets) is a crucial ingredient for our analysis. That said, there are only a few places we use the finiteness of the label space, and we suspect an even more general result could go through under some regularity assumptions.
> 2. Thank you for pointing out this source of confusion. Please see our response to reviewer yteg for the general point about positive vs negative results from Finocchiaro et al. [8].  On the specific subject of the Lovasz hinge, [8] shows that the surrogate is consistent for a variant of the original target problem, where you are allowed to abstain on a subset of labels. What we actually intended to write is that Theorem 1 would apply to Lovasz hinge and this “set-valued abstain” problem that it is consistent for. We will add this clarification to the paper.
>
> Finally, thank you for taking the time to compile these additional references!  We will certainly include them.

---

### Official Review · Reviewer_J1xj · 2021-07-19

**Rating:** 5
**Confidence:** 4

**Summary:**

This paper shows the existence of linear regret bounds for polyhedral surrogates. While such bounds have been proven for specific surrogates this paper shows any calibrated linear surrogate has linear regret bounds. In addition it also shows any surrogate that is essentially not piecewise linear has only quadratic regret bounds, i.e. there exists distributions where target-regret >= sqrt(surrogate-regret).



**Ethical Concerns:**

None.

**Limitations And Societal Impact:**

None.

**Main Review:**

The paper puts together several results in a convenient abstract form for future research. However, there is no surprise factor in any of the results. Both linear regret bounds for piecewise linear surrogates, and a quadratic lower bound for "strongly-convex" surrogates would not surprise researchers familiar with the area. The abstract proof techniques and definitions however are interesting and are the main contribution of the paper.

Other comments:

1. There are several properties whose implications are not clear. e.g. some of the properties are
A. L is polyhedral.
B. L,psi are consistent
C L,psi are calibrated
D. L,psi indirctly elicit l
E. psi is epsilon separated w.r.t. Gamma

Fact 1 says B implies C imples D. But there are other implications that are missed. e.g. C also implies B. (Tewari et al. (2007), Ramaswamy 2016). I am also guessing A and D implies C. A figure illustrating all these properties using a block diagram would be useful.

2. The polyhedral losses definition seems to exclude non-convex polyhedral losses. Is that on purpose?
3. The separated link definition (informal version) seems wrong. e.g. with hinge loss surrogate and sign predictor for the binary zero-one loss, psi(0.001) is an optimal prediction but psi(-0.001) is not for some p but |0.001+0.001| is not greater than epsilon for this case.
4. All these concepts would be better served by a more simple example, like the Crammer-Singer surrogate than the significantly more complex BEP surrogate.

--- After rebuttal ---

I thank the authors for their responses. I have read the other reviews and replies, and hence revise my score to 5 from 4. I still think a large portion of these results by themselves are not exciting/interesting. But this paper does provide the language and tools required for such results.



**Time Spent Reviewing:**

6

---

> ### Author Response · Authors · 2021-08-10
> **Response: surprise and other comments**
>
> **Surprise:** We do find the results surprising in their generality, both in that they are true (e.g. one gets linear transfer rates for *any* discrete target loss and *any* consistent polyhedral loss + link) and that it is possible to prove them: consider the large amount of work devoted in papers like [18; section 8, pp 14-23] in proving linear transfers for particular closed-form surrogates relying on details of the function. Moreover, even if not surprising to the reviewer, we believe the bounds are nonetheless useful, as we describe in response to reviewer yteg.
>
> **Other comments:**
> 1. Thank you for the suggestion! In fact, A and D do not alone imply C, but {A,D,E} do imply C. To see that A and D do not suffice, consider hinge loss where instead of linking (-infty,0) to - and [0,infty) to +, you link (-infty,1) to - and [1,infty) to +. A and D are satisfied. Yet C will fail because if a distribution puts all weight on +, one can still approach 0 loss while linking to -. This point is similar to your point 3 below.
> 2. Yes, correct. We use “polyhedral” precisely because it typically means “piecewise-linear and convex” (and a finite number of pieces); at least we are not aware of any usage of “polyhedral” that allows for non-convex functions or sets.
> 3. Thank you and nice catch: while the formal definition is correct, the informal definition above it was not. The informal version should instead say “when u* is an optimal surrogate prediction and psi(u) is not an optimal target prediction, …”.
> 4. Thank you for the suggestion; we will add a few simple examples.

---

### Official Review · Reviewer_yteg · 2021-08-01

**Rating:** 6
**Confidence:** 3

**Summary:**

The paper concerns theory of surrogate losses. The authors present general results concerning surrogate regret bounds which characterize how fast the target loss approaches the best possible solution if a surrogate loss is optimized. The authors point out that families of surrogate losses are characterized by different generalization rates (how fast the surrogate loss approaches optimum as a function of the number of training points) and regret transfer rates (how the surrogate regret translates to target regret). The authors focus on polyhedral losses (piecewise-linear convex loss functions; hinge loss is an examples of such function). Particularly, they show that the regret transfer for consistent polyhedral surrogates (a surrogate loss is consistent if its all optimal solutions are also optimal for the target loss) is linear (which is a desired rate). On the other hand, they show that for "sufficiently non-polyhedral" surrogates (like logistic loss) the transfer rate is no faster than a square root. Nevertheless, the generalization rate for smooth losses can be as fast as $\frac{1}{n}$, while for polyhedral losses is $\frac{1}{\sqrt{n}}$. All together it leads to comparable surrogate regret rates.




**Ethical Concerns:**

No ethical concerns.

**Limitations And Societal Impact:**

The paper advances understanding of learning theory. There are no negative societal impacts.

**Main Review:**

This is a purely theoretical paper without any empirical evidence. It reads well, particularly the first part of the paper. It states and tries to answer an important question of regret rates for surrogate losses.

### Originality

The question stated in the paper is one of the most important in theoretical machine learning. However, its practical implications can be of less importance as usually regret bounds are very loose. Nevertheless, they can serve as sanity checks and can be treated as guidelines for development of learning algorithms. The presented results nicely generalize previous findings. They are not so surprising, but give almost exhaustive picture of surrogate loss minimization.

### Quality

Unfortunately, I was not able to check the appendix. Nevertheless, the theoretical results sound plausible. However, the paper has several drawbacks that decrease its final evaluation:

- It is not clear what is the final goal of the paper. I suppose that the authors aim at comparing polyhedral and smooth surrogate losses. Unfortunately, I am missing examples of bounds for a variety of problems for both types of losses. An experimental comparison would also be desired, as authors write that the results will be useful for practitioners.

- It is also not clear to what machine learning problems the results apply. It is clear that for binary classification under 0/1 and abstain loss. But what are the other problems? The authors say that one can construct consistent polyhedral surrogates for any discrete target loss. But this general sentence is hiding important message for practitioners. The results in [8] show that there are important practical problems for which polyhedral losses are inconsistent and therefore the presented findings in this paper do not apply to. An example of such problem is top-$k$ classification. The paper should discuss in more detail what usual problems are "positive" and "negative" instances of the presented theory.

### Clarity

This is a purely theoretical paper, but the authors made an effort to make it easy to follow, especially the first part. The second part is much more dense (because of the page limit) therefore requires much more time to go through. To improve clarity, I would encourage authors to give more examples of problems and losses to which the theory applies. Also, a comparison of the bounds for polyhedral and smooth surrogates would make the final message much clearer.

Minor comments concerning clarify are given below:

- Weak optimality: What is the meaning of "weak optimal" in the considered case? This statement appears twice in the paper, but there is no definition or interpretation of it given.

- Link function: How often the name link function is used in this context? In statistics, link function has a very precise definition. There is some analogy to the context used by the authors, but a different name such as prediction or inference function would probably sound better and less confusing. Let us consider binary classification and logistic regression for which we have: real scoring function, link function that maps real values to probabilities, and finally prediction function that translates probabilities to binary predictions. The other names are also commonly used in multi-label classification or structured-output prediction where procedures of translating surrogate scoring functions to final predictions are usually quite complex.

### Significance:

In general, the results given in the paper are of high importance for theory of machine learning. Nevertheless, the significance is decreased by the lack of the final comparison between polyhedral and smooth losses and detailed discuss about the problems to which the theory does and does not apply.



**Time Spent Reviewing:**

6

---

> ### Author Response · Authors · 2021-08-10
> **Response**
>
> **Goal of the paper:** We view our goal as two-fold: 1. To prove a broad surrogate regret bound for all polyhedral surrogates; 2. To separate the transfer rates of polyhedral and smooth surrogates, as part of a broader agenda of deepening our theoretical understanding of surrogate loss design. As discussed later in the response, the linear transfer rate from goal 1 immediately applies to a number of existing polyhedral surrogates of interest to practitioners, some of which did not have known transfer rates.
>
> **Positive vs negative instances:** We would like to clarify the relationship to [8]. While [8] states that some polyhedral surrogates are not consistent for their desired targets (e.g. the Lapin et al. surrogate for top-k, and the Lovasz hinge), it also provides positive results for these surrogates/problems. Namely, (a) for any given surrogate, even if it is not consistent for the desired target, it is consistent for some other target (the one it “embeds”), and (b) given any desired target, there exists a consistent polyhedral surrogate for it. Our results apply in both cases. For (a), the commonly-used polyhedral surrogates have linear transfer rates for the problems they are consistent for, meaning the Lapin et al. surrogate and Lovasz hinge have linear rates for the problems they actually solve. For (b), the desired targets necessarily have some polyhedral surrogates with linear transfer rates. We thank the reviewer (along with reviewer auqc) for pointing out this source of confusion; we will clarify in the paper as well.
>
> **Weakly optimal:** By “weakly optimal” we mean that the overall generalization rate to the target problem has the optimal dependence on the number of data points. We will clarify.
>
> **Link function:** Thank you for the comment. While “link function” is a specific function for generalized linear models (perhaps what the reviewer is referring to?), and some machine learning papers avoid the term and opt instead for “prediction function”, many papers do use “link function”.  For example, from a quick browse through our references the papers [1,8,9,12,13,15,19,21] use “link function”, among others.  Because we are building on this same body of work, we prefer to use the same terminology, but will certainly clarify our meaning.

---

> > ### Comment · Reviewer_yteg · 2021-09-07
> > **After authors' response**
> >
> > I thank the authors for their responses.
> >
> > The paper would be greatly improve if more examples of standard problems satisfying the theoretical results were given. The existence results are certainly interesting, but giving more concrete examples will make the story more complete. The authors do not clearly discuss limitations of their results and the reader needs to guess (i.e., carefully analyze the setting and the results) to what real problems the theory applies.
> >
> > Taking the above into account I will stay with my original evaluation.

---

### Decision · Program_Chairs · 2021-09-27

**Decision:**

Accept (Poster)

**Comment:**

The paper shows that every consistent polyhedral surrogate loss has a linear surrogate regret bound, while if the loss is sufficiently non-polyhedral (strongly convex) the regret bound is a square root (fast surrogate generalization rates translate to slow rates).

The reviewers liked the generality of these results and most of them believe this is a solid contribution to a general theory of surrogate losses. The technical part of the paper is sound. Clearly, the paper answers an interesting open question.

While I recommend the paper for acceptance, I still have two concerns:
- The results are based on the definition of the regret as the suboptimality with respect to the optimal prediction function **among all functions**, rather than optimal **within some class of functions** (e.g., linear). So these results are only useful if the learning method is able to converge to the optimal predictor at all (i.e. either the optimal is within the class of functions the method works with, or the method is universal). I guess this criticism applies to a general research topic on the consistency of surrogate losses rather than specifically to this particular paper.
- While having linear regret transfer is definitely nice, losses without curvature (polyhedral) generally suffer from the slow rates of convergence (w.r.t. sample size) $O(1/\sqrt{n})$ in the worst case, while strongly convex losses have fast rates $O(1/n)$. So after all, it is not clear which losses work best.